# A Bronze-Tomato Enriched Diet Affects the Intestinal Microbiome under Homeostatic and Inflammatory Conditions

**DOI:** 10.3390/nu10121862

**Published:** 2018-12-02

**Authors:** Marina Liso, Stefania De Santis, Aurelia Scarano, Giulio Verna, Manuela Dicarlo, Vanessa Galleggiante, Pietro Campiglia, Mauro Mastronardi, Antonio Lippolis, Mirco Vacca, Anastasia Sobolewski, Grazia Serino, Eugenio Butelli, Maria De Angelis, Cathie Martin, Angelo Santino, Marcello Chieppa

**Affiliations:** 1National Institute of Gastroenterology “S. de Bellis”, Institute of Research, 70013 Castellana Grotte (BA), Italy; marinaliso@libero.it (M.L.); giu.verna@gmail.com (G.V.); manueladicarlo@alice.it (M.D.); vanessa.galleggiante@libero.it (V.G.); mauro.mastronardi@irccsdebellis.it (M.M.); antonio.lippolis@irccsdebellis.it (A.L.); grazia.serino@irccsdebellis.it (G.S.); 2Department of Pharmacy, School of Pharmacy, University of Salerno, 84084 Fisciano (SA), Italy; s.desantis.expimmunopathol@gmail.com (S.D.S.); pcampiglia@unisa.it (P.C.); 3Institute of Sciences of Food Production C.N.R., Unit of Lecce, 73100 Lecce, Italy; aurelia.scarano@ispa.cnr.it (A.S.); angelo.santino@ispa.cnr.it (A.S.); 4Department of Soil, Plant and Food Sciences, University of Bari, 70126 Bari, Italy; mirco.vacca.mv11@gmail.com (M.V.); maria.deangelis@uniba.it (M.D.A.); 5School of Pharmacy, University of East Anglia, Norwich Research Park, Norwich NR4 7TJ, UK; a.sobolewski@uea.ac.uk; 6John Innes Centre, Colney Research Park, Norwich NR4 7UH, UK; eugenio.butelli@jic.ac.uk (E.B.); cathie.martin@jic.ac.uk (C.M.)

**Keywords:** microbiota, inflammatory bowel disease (IBD), polyphenols, bronze tomatoes, murine models

## Abstract

Inflammatory bowel diseases (IBD) are debilitating chronic inflammatory disorders that develop as a result of a defective immune response toward intestinal bacteria. Intestinal dysbiosis is associated with the onset of IBD and has been reported to persist even in patients in deep remission. We investigated the possibility of a dietary-induced switch to the gut microbiota composition using Winnie mice as a model of spontaneous ulcerative colitis and chow enriched with 1% Bronze tomato. We used the near isogenic tomato line strategy to investigate the effects of a diet enriched in polyphenols administered to mild but established chronic intestinal inflammation. The Bronze-enriched chow administered for two weeks was not able to produce any macroscopic effect on the IBD symptoms, although, at molecular level there was a significant induction of anti-inflammatory genes and intracellular staining of T cells revealed a mild decrease in IL17A and IFNγ production. Analysis of the microbial composition revealed that two weeks of Bronze enriched diet was sufficient to perturb the microbial composition of Winnie and control mice, suggesting that polyphenol-enriched diets may create unfavorable conditions for distinct bacterial species. In conclusion, dietary regimes enriched in polyphenols may efficiently support IBD remission affecting the intestinal dysbiosis.

## 1. Introduction

The link between the intestinal microbiota and intestinal chronic inflammatory syndromes has been increasingly investigated over the past 20 years. Alterations in the composition of gastrointestinal microbiota (gut dysbiosis) have been associated with numerous conditions, including metabolic disease and inflammatory bowel disease (IBD) [1,2].

Due to the crucial involvement of the microbiota in onset of IBD, strategies targeting IBD-associated intestinal dysbiosis are considered among the most promising therapies for future treatment of IBD patients [3,4]. IBDs include two major disorders: ulcerative colitis (UC) and Crohn’s disease (CD), both characterized by chronic intestinal inflammation and relapsing clinical courses, but with distinct anatomic localization and inflammatory mediator profiles [1,2]. The triggers of IBDs are as yet unknown, however, it is likely the origins are multifactorial resulting from a combination of different factors that include genetic predisposition (such as NOD2 mutations) [5,6,7,8] and environmental components such as childhood exposure to antibiotics [9,10,11] and diet [12,13,14,15]. These components seem to perturb the composition of the intestinal microbiota promoting a switch from the “healthy” microbiota to the “pathological” condition of dysbiosis [16,17,18].

We recently investigated the protective effects mediated by the administration of a diet enriched with a tomato line called Bronze [19]. This near isogenic tomato line is enriched in several different polyphenols and proved to be a powerful and innovative tool to evaluate the effects of plant bioactive compounds, particularly in the context of the intestinal immune response in IBD [19]. We demonstrated that the administration of a nutritional regime enriched with 1% of Bronze tomato fruit was able to promote a change in the composition of the microbiota in healthy mice and partially suppress the host inflammatory response to reduce/delay the appearance of intestinal damage induced by DSS administration [19].

In contrast to the previous experimental setup, we built on our published findings and used a murine model of spontaneous UC, Winnie, to investigate the therapeutic rather than the preventive potential of the Bronze-enriched diet. Winnie mice are characterized by a point mutation in the Muc2 gene that induces endoplasmic reticulum stress, intestinal barrier dysfunction and ultimately an UC-like disease. Winnie mice develop chronologically progressive IBD symptoms resembling human UC [20], including erosion of intestinal epithelium, increased intestinal cellularity, neutrophil recruitment and appendiceal microbiome dysbiosis. Our aim was to determine the therapeutic potential of a dietary intervention based on the Bronze-enriched diet in adult Winnie mice. 

Our results indicate that, despite undetectable morphological improvement, two weeks of Bronze-enriched diet was able to elicit the expression of several anti-inflammatory related genes both in Wild type (WT) and Winnie mice. Furthermore, the bronze-enriched diet significantly impacted the intestinal microbiome of both WT and Winnie mice.

## 2. Materials and Methods

### 2.1. Ethics Statement

Our investigations were conducted in accordance with national and international guidelines and were approved by the authors’ institutional review board (Organism For Animal Wellbeing—OPBA). All animal experiments were carried out in accordance with Directive 86/609 EEC enforced by Italian D.L. n. 116/1992, and approved by the Committee on the Ethics of Animal Experiments of Ministero della Salute–Direzione Generale Sanità Animale (Prot. 768/2015-PR 27/07/2015) and the official RBM veterinarian. Animals were sacrificed if found to be in a severe clinical condition, to avoid undue suffering.

### 2.2. Generation of Tomato Lines and Diets

The Bronze tomato line (*E8:MYB12*, *E8:Del/Ros*, *35S:StSy*) was developed as previously described [19].

### 2.3. Murine Models

Sex- and weight-matched mice were divided into four groups (four mice each). WT mice were purchased from Jackson Laboratories: (C57BL/6, Stock No. 000664), while Winnie mice were obtained from the University of Tasmania.

Mice, pellet consumption and drinking water were monitored on a daily basis. Each group of mice received a different diet. Freeze-dried tomato was supplemented by addition to a standard rodent diet (4RF18) at 1% (tomato based-diets). Groups of mice were fed with the different tomato supplemented diets for two weeks. Body weight, stool consistency and rectal bleeding were recorded. Mice were sacrificed at day 14, and colon and mesenteric lymph node (MLN) tissues were explanted to evaluate the clinical severity of colitis. Colon length was measured as an indicator of colonic inflammation. The colon/body weight indices were calculated as the ratio of the colon wet weight and the total body weight (BW), and as the ratio of the colon length and the total BW of each mouse. Body weight, occult and rectal bleeding and stool consistency were monitored daily. Disease activity index (DAI) was determined by scoring changes in body weight, occult blood and gross bleeding.

### 2.4. DNA Extraction from Stool

Total genomic bacterial DNA was isolated from frozen stool samples of mice using the QIAamp® Fast DNA Stool Mini Kit (QIAGEN, Hilden, Germany), according to the manufacturer’s instructions. 

### 2.5. Bacterial Microbiome Estimated by 16S rRNAs Metagenetics

16S metagenetic analyses were carried out at Genomix4life (spin-off of the University of Salerno, Fisciano, Italy) by using the Illumina MiSeq platform. The V3-V4 region of the 16S rRNA gene was amplified for analysis of diversity inside the domains of Bacteria [21]. PCR and sequencing analyses were carried out according to the protocol of Genomix4life. Quality control (QC) and taxonomic assignments were undertaken according to the QIIME and the Ribosomal Database Project Bayesian classifier in combination with a set of custom designed informatics pipelines implemented by Genomix4life for analyses of microbial communities. Taxonomic attribution was carried out using the BLAST search in the NCBI 16S ribosomal RNA sequences database [22]. The percentage of each bacterial OTU was analyzed individually for each sample, providing relative abundance information among the samples based on the relative numbers of reads within each [23]. Alpha diversity (observed species, Chao1 richness and Shannon diversity indices) was calculated using Qiime [24,25]. Differences in microbial communities between mouse groups were also investigated using the phylogeny-based, unweighted Unifrac distance metric [26].

### 2.6. Cytofluorimetric Analysis

FoxP3 staining: Mesenteric lymph nodes (MLNs) were isolated from mice fed with tomato (Control or Bronze)-enriched food. MLNs were passed through a 30 m cell strainer (Miltenyi Biotec, Bergisch Gladbach, Germany) to obtain a single cell suspension and then washed with DPBS (Gibco, Waltham, MA, USA) + 0.5% bovine serum albumin (BSA, Sigma-Aldrich, St. Louis, MO, USA). Single cell suspensions were stained with CD4-FITC and CD25-PE (Miltenyi Biotec, Bergisch Gladbach, Germany). Cells were then permeabilized with Foxp3 Fixation/Permeabilization Kit (eBioscience, San Diego, CA, USA) and subsequently washed with PERM Buffer (eBioscience, San Diego, CA, USA). Finally, cells were stained with Foxp3-APC (Miltenyi Biotec, Bergisch Gladbach, Germany), according to the manufacturer’s instructions. Flow Cytometer acquisition was performed using NAVIOS (Beckman Coulter, Brea, CA, USA).

T cell Intracellular Staining: T cells from MLNs of mice fed with tomato (Control or Bronze)-enriched food were cultured with a 500X Cell Stimulation Cocktail (eBiosceince, San Diego, CA, USA) for 12 h, washed with DPBS + 0.5% BSA and stained with CD4-APC-Vio700 (Miltenyi Biotec, Bergisch Gladbach, Germany). After washing, cells were then permeabilized with BD CytoFix/CytoPerm^®^ Fixation/Permeabilization Kit^®^ (BD Biosciences, Franklin Lakes, NJ, USA), washed with PERM Buffer, and stained with: IL-17A-FITC, TNF-PE and IFN-APC according to manufacturer’s instructions (Miltenyi Biotec, Bergisch Gladbach, Germany). Flow Cytometer data analysis was performed using NAVIOS (Beckman Coulter).

### 2.7. RNA Extraction and qPCR Analysis

Total RNA was isolated from the medial colon of WT and Winnie mice. The RNA was extracted using TRIzol^®^ (Thermo Fisher Scientific, Waltham, MA, USA) according to the manufacturer’s instructions. Total RNA (1 g) was reverse transcribed using the iScript cDNA Synthesis kit (Biorad, CA, USA) with random primers for cDNA synthesis. Gene expression of Tnf-, Ifn, Il-10, Hmox1, Slpi and Gapdh was assessed using the TaqMan gene expression assay (Thermo Fisher Scientific, MA, USA) murine probes: Mm00443258_m1, Mm01168134_m1, Mm00439614_m1, Mm00516005_m1, Mm00441530_g1 and Mm99999915_g1, respectively. Real-time analysis was performed on a CFX96 System (Biorad, CA, USA) and for the relative expression the ΔΔCt method was used.

### 2.8. Morphological Analysis

Tissue sections from the distal colon were fixed in 10% buffered formalin and embedded in paraffin. Sections of 3 μm were stained using a hematoxylin eosin standard protocol. Images were acquired using Leica LMD 6500 (Leica Microsystems, Wetzlar, Germany).

### 2.9. Statistical Analysis

All data were expressed as the means ± SEM. All results were obtained from three consecutive and independent experiments. 

Metagenomic data (Unifrac distance metric and taxonomic abundance) were analyzed by Principal Component Analysis (PCA) [27,28] using a statistical software Statistica for Windows (Statistica 6.0 for Windows 1998, StatSoft, Vigonza, Italia). Samples more similar to each other should appear closer together according to the respective axis reflecting the variation among all samples [29]. This technique is useful for displaying clusters existing within data. The variables (features) reflect the relative bacterial composition in a sample at a particular taxonomic level. In addition, Permut-MatrixEN software was used to identify clusters at the level of the mouse groups and taxa [30]. 

Statistical analysis of the relative abundances of microbial genera was based on Duncan’s Multiple Range test, with a significance level of *P* ≤ 0.05. Finally, unless specifically described, other data and group differences were analyzed and compared by paired or unpaired, two-tailed Student’s *t*-tests. Grouped analyses were performed with the two-way ANOVA test, using Bonferroni as a post test.

## 3. Results

### 3.1. Administration of a Bronze Tomato-Enriched Diet to the Murine Model of Ulcerative Colitis, Winnie

We recently demonstrated that the administration of a murine chow enriched with 1% of dried Bronze tomato was able to prevent/delay the intestinal damage induced by DSS administration using WT C57Bl/6 mice [19]. However, we did not determine whether the Bronze tomato dietary supplement was able to exert beneficial effects in the context of an ongoing intestinal inflammation. Here, we explored the effects of nutritional intervention in a model of spontaneous progressive ulcerative colitis. WT and Winnie mice were fed for 2 weeks with 1% control (Control) or bronze (Bronze) tomato-enriched diets. Figure 1A shows the experimental setup. Administration of the enriched diets did not alter the weight of WT and Winnie mice (Figure 1B) nor their colon weight and length (Figure 1C,D, respectively). Morphological analysis of the explanted colon did not reveal differences between Control and Bronze treated mice (Appendix A).

### 3.2. IL-17A Reduction in CD4^+^ Mesenteric Lymph Node (MLN) T Cells Treated with a Bronze Tomato Diet

Following two weeks of tomato-enriched diets, we isolated and analyzed the intracellular cytokine production of the MLNs CD4^+^ T cells. The Bronze enriched diet resulted in a reduced percentage of CD25^+^Foxp3^+^ (Figure 2A) and increased percentage of TNFα^+^ cells (Figure 2B) compared to the Control diet. Both changes were not significantly different, but the trend was consistent independent of the genotype of the mice. A Bronze diet resulted in a significant reduction in the percentage of IL-17A CD4^+^ cells in WT mice and a similar trend in Winnie mice, while the percentage of IFNγ^+^CD4^+^ cells was unchanged (Figure 2C) both in WT and Winnie. Of note, the percentage of IFNγ^+^CD4^−^ cells was reduced by the Bronze diet, particularly in Winnie mice (Figure 2D).

### 3.3. Bronze Tomato Diet Molecular Signature

We previously demonstrated that the administration of a Bronze enriched diet was able to reduce the expression of inflammatory cytokines (Il-6, Tnf-α, Il-1α, Il-1β, Il-12) in the colon of WT mice [19] and that the expression level of Slpi was a reliable marker for the oral intake of quercetin [31,32]. We next analyzed the expression levels of Tnf-α, Ifnγ, Il-10, Slpi and Hmox1 in the colon of WT and Winnie mice. While both a Bronze and Control enriched diet significantly increased Il-10 expression in WT mice (Figure 3A), only a Bronze diet induced a higher increase in Il-10 in Winnie mice. Tnf-α and Ifnγ colon expression was generally lower in mice fed with Control or Bronze diets in both the Winnie and WT mice (Figure 3B,C). A significant reduction in Ifnγ expression was observed in Winnie mice fed with a Bronze diet compared to Day 0, but not in Winnie mice fed with a Control diet. Slpi, a major checkpoint for the anti-inflammatory activity of quercetin [31,32], was significantly induced by both the Control and the Bronze-enriched diets both in WT and Winnie mice (Figure 3D). Slpi induction was higher in Winnie compared to WT mice. Hmox1 expression was similar to that of Slpi, suggesting an anti-inflammatory pathway that is induced by the Bronze diet (Figure 3E).

### 3.4. Dysbiotic Intestinal Microbiota Communities Changed Following Two Weeks on a Bronze Tomato Diet in Winnie Mice

Time zero for each mouse was collected both for WT and Winnie mice. At day 0 for the mice fed with Control tomato-enriched diet, the Shannon index of the Winnie mice was higher than the WT mice (2.413 and 2.180, respectively, *P* = 0.011). No statistically significant differences were found for the number OTUs and Chao index (Appendix A). To investigate the response of the microbiota to the host intake of different dietary regimes, fecal material was collected from mice following two weeks of Bronze or Control enriched chow. After two weeks of feeding Control tomato, the WT mice increased their OTUs and their Shannon index compared to Day 0. Compared to the Control tomato diet, all WT and Winnie mice showed non-significant reductions in OTUs on the Bronze tomato diet. In our previous studies, the diversity and richness of the microbiota of mice did not show significant differences (*P* > 0.05) in response to different diets [19]. The 3 phylogeny-based β-diversity analyses did not show statistical separation between the composition the microbiome of fecal mouse samples after two weeks of Bronze or Control tomato diets (Appendix A). Samples were mainly clustered according to the genotype (WT or Winnie mice). 

The evaluation of the 16S revealed a significant difference (*P* < 0.05) between 20-week old WT and Winnie mice. In particular, the relative abundance of *Bacteroidetes* and *Firmicutes* (41.97% and 39.26% in WT vs. 60.62% and 23.36% in Winnie mice), as well as the ratio between them (1.14 vs. 2.76 respectively, *P* = 0.022) confirmed differences already known for hosts characterized by ongoing intestinal inflammation [33,34,35,36]. Two weeks of dietary intervention changed the microbial content of both WT and Winnie mice. The differences in bacterial abundance monitored by 16S segregation, observed at time zero, were later lost as fecal material from WT and Winnie mice could not be clearly differentiated (Figure 4). 

We aimed to identify bacterial genera susceptible to the polyphenol-enriched diet. We compared Control and Bronze-fed groups independently of genotype (Figure 5). Data revealed that two weeks of Bronze diet resulted in an increase of *Flavobacterium* and a reduction in the relative abundance of *Oscillospira*. When considering mice in relation to their different genotypes, the relative increase in specific bacterial content was directly related with permissive diet. Genotype-specific differences appeared as a direct consequence of the starting concentration of specific bacteria before the dietary intervention. Particularly, with the Bronze diet we detected a significant increase in *Lactobacillus* and *Parabacteroides* in WT mice. The same trend was observed for *Parabacteroides*, but did not reach significance in Winnie mice, which were characterized by an increase in *Odoribacter*. An opposite trend was observed for *Desulfovibrio* and *Natronincola* that decreased in WT mice following Bronze diet administration, while *Akkermansia* and *Blautia* decreased significantly in Winnie mice. 

At the species level, WT mice showed a different profile compared to Winnie mice (Appendix A). The Bronze-enriched diet was able to drive the changes in bacterial species in both WT and Winnie mice. Data revealed that two weeks of Bronze diet resulted in an increase of the relative amount of *Akkermansia muciniphila* in both WT and Winnie mice compared to the Control tomato diet (Appendix A). In contrast, the Bronze- enriched diet drove a reduction in the relative content of *Bacteroides* species (*B. chinchilla, B. rodentium, B. xylanisolvens*), *Escherichia albertii, Parabacteroides distasonis* and *Ruminococcus gnavus* in both WT and Winnie mice. 

## 4. Discussion

IBD includes Crohn’s disease (CD) and ulcerative colitis (UC), with idiopathic chronic inflammation of the intestine often occurring at a young age. Treatment of IBD has changed since the introduction of monoclonal antibodies blocking the inflammatory cytokine TNFα have augmented traditional treatments, such as corticosteroids and thiopurines. Alternative biological intervention is extremely useful in case of early-stage treatment of patients with IBD forms resistant to conventional therapy [37,38,39]. Together with the introduction of numerous new therapeutic strategies, the assessment of the status of IBD activity is being addressed. Terms like “mucosal healing”, “biochemical remission” and “deep remission” are now commonly used to discriminate between different degrees of response to the therapeutic intervention of patients [40]. Mucosal healing is an endoscopic assessment of the absence of inflammation, nonetheless, it is possible to have mucosal healing in patients with elevated fecal levels of calprotectin indicating an ongoing process of neutrophil infiltration. It is not surprising that mucosal healing is not a good predictor of the desired sustained clinical remission [41]. Furthermore, dysbiosis has been reported in IBD patients, even in the absence of inflammation [42], suggesting that unless corrected, the dysbiosis may play a role in disease recurrence. Finally, the reduced diversity in the microbiota of UC patients [43] has a negative influence on the production of metabolites, such as butyrate, that sustain intestinal epithelium integrity [44]. Indeed, butyrate producing bacteria are scarcely represented in the microbiota of UC patients [43], suggesting that intervention to correct dysbiosis should be provided before dietary supplementation that support butyrate production (such as soluble fibers).

We recently described the effects of a diet supplemented with a tomato line enriched in different polyphenols [19] as an innovative tool to evaluate beneficial effects of plant bioactive compounds on the disease activity index in a mouse model of IBD. We also demonstrated that the administration of a murine chow enriched with 1% of Bronze tomato fruit was able to affect numerous biological features of the intestinal immune response in WT mice. Furthermore, we also showed that two weeks of the Bronze nutritional regime profoundly changed the composition of the intestinal microbiota of WT mice compared with the administration of diets enriched with near isogenic tomato lines [19]. This present study aimed to identify whether a Bronze diet could exert beneficial effects during inflammation. In order to address this, the mouse model Winnie was used as its phenotype resembles the progressive intestinal inflammation observed in human UC. 

Two weeks of nutritional intervention with Bronze-enriched diet was not able to correct the macroscopic features observed in 20-week old Winnie mice, which included weight loss, colon shortening and loss of stool consistency. Surprisingly, at the intracellular level, it was possible to detect a decrease in the percentage of cells positive for CD4^+^IL17A and CD4^−^IFNγ in the MLNs. Analysis of gene expression in the colonic tissue revealed a significant induction of Slpi both in Winnie and WT mice, confirming previous observations of bone marrow derived dendritic cells exposed to quercetin [31,32]. Slpi is a potent NF-kB inhibitor, its expression and secretion are able to suppress inflammation and facilitate tissue repair. Accordingly, the Bronze-enriched diet was able to induce the expression of Hmox1, another potent anti-inflammatory protein recently recognized as an important checkpoint for intestinal inflammation [45]. Both tomato-enriched diets induced Il-10 expression in colonic tissues, but the Bronze-enriched diet was still able to further promote Il-10 expression compared to Control tomato diet in the UC model Winnie. The Bronze-enriched diet was able to reduce Ifnγ and Tnf-α in both WT and Winnie mice, although a significant reduction was evident only for Ifnγ in Winnie mice. Similar trends were observed for Control tomato-enriched diets, although the reductions in expression did not reach significance suggesting that the polyphenol-enriched component of the Bronze tomatoes was, at least partially, involved in reduction of Ifnγ and Tnf-α expression. These data were in line with the observations in MLNs. 

The potential of nutritional intervention, enriched with bioactives, to modulate an already established intestinal inflammation has implications for the development of future treatments for IBD. The intestinal microbiota and its dysbiosis are thought to play a fundamental role in either the cause, persistence or recurrence of IBD [46,47]. 

Strikingly, the Winnie mice were characterized by differences in their intestinal microbial content compared to WT mice (Appendix A). The administration of the Bronze-enriched diet was also able to reshape the microbial community of both genotypes, WT and Winnie. We recently explored the possibility that some polyphenols, including quercetin in particular, work as an iron-chelating agent [48]. When nutritional regimes enriched in quercetin, such as the Bronze diet, are introduced into the diet, iron sequestration may create challenging conditions in the intestinal lumen. This mechanism is equally true for homeostatic and inflammatory conditions, thus the final results may partially overlap. Some microbial species may be able to resist iron starvation better, while others may proliferate less and become underrepresented. In the present study we obtained some results showing genotype-independent trends for decreases in *Oscillospira* and increases in *Flavobacterium* even though of different magnitudes. On a genera level, differences in the responses of communities to the Bronze diet were less consistent between WT and Winnie mice. This result could be related to the number of bacteria represented at the beginning of the dietary intervention, where species underrepresented at time zero, will likely fail to receive benefits from the polyphenol-enriched diet compared to the predominant microbiota present at the beginning of the intervention. 

## 5. Conclusions

The results from the present study will be important for understanding the effects of nutritional intervention as an adjuvant in the treatment of IBD [49,50,51,52]. 

Nutritional regimes fortified in bioactive compounds may help to correct intestinal dysbiosis, prevent recurrence and support complete remission of IBD. More studies will be required to design the most effective strategies to be used, particularly for patients affected by intestinal dysbiosis. The diet may create favorable conditions for healthy bacterial growth, but may prove ineffective in the absence of concomitant probiotic administration. Furthermore, using different experimental strategies, it will be crucial to assess efficiency and safety of a prolonged administration to a Bronze-enriched dietary regime. Will the microbiota community become stable and the inflammatory response been suppressed? Is the microbiota community Bronze-dependent and the dietary intervention be suppressed once stabilized? We still largely ignore numerous aspects of the axis between nutrition–inflammation and microbiota selection. With the present results, we pave the way for future studies focused on bioactive- enriched food acting as adjuvants for IBD patients in deep remission from dysbiosis to prevent disease recurrence.

## Figures and Tables

**Figure 1 nutrients-10-01862-f001:**
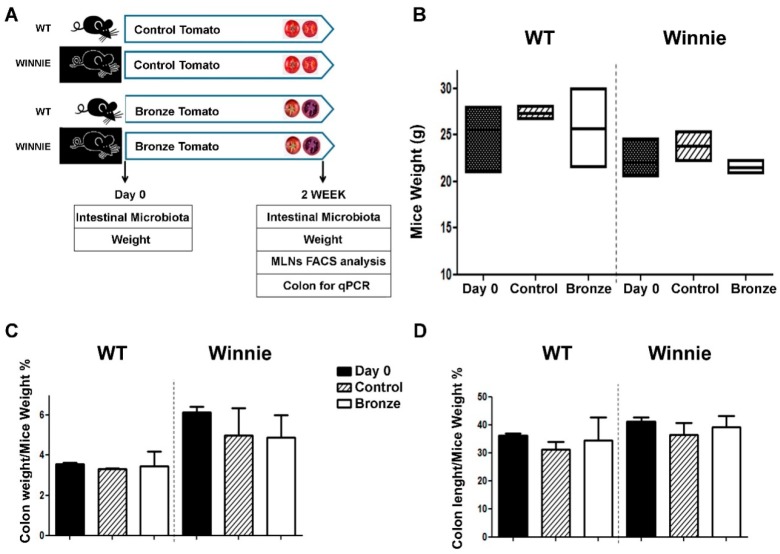
Experimental design and macroscopic characterization of the experimental groups. (**A**) Sex-matched mice were divided into four groups based on their genotype (Wild Type or Winnie) and their diet (enriched with Control or Bronze lyophilized tomato fruit). Mouse weight was recorded at the beginning of diet administration (Day 0) and at the end of the trial (2 Weeks). Fecal samples were collected for microbial meta-analyses at both time points (Day 0 and Week 2). Tissues were explanted and analyzed at the end of the treatments as indicated. Analysis of mice from different groups included: mouse weight (**B**), colon weight/mouse weight (**C**) and colon length/mouse weight (**D**). Black bars show the values at Day 0, striped bars show the Control- and white bars the Bronze-enriched diet. Statistical evaluation was performed using unpaired two-tailed Student’s *t*-tests.

**Figure 2 nutrients-10-01862-f002:**
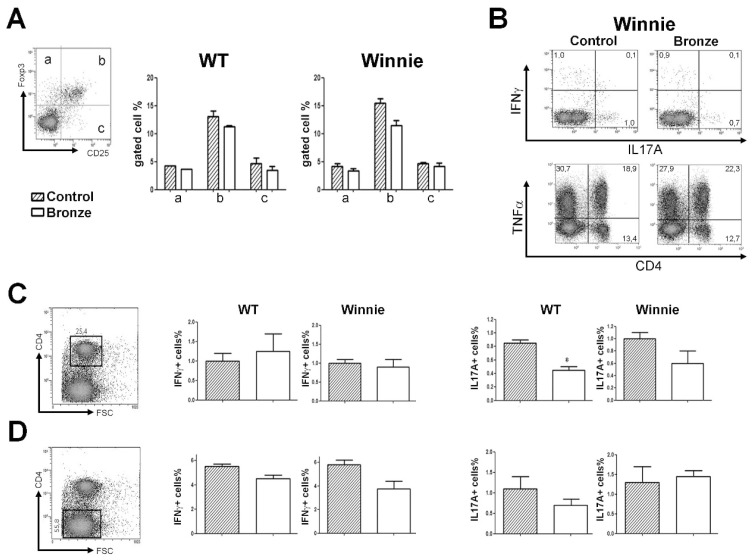
Mesenteric lymph node T cell cytokine staining. (**A**) Representative Treg staining of CD4+ cells. Histograms represent the percentages of Foxp3^+^CD25^+^ cells in the MLNs of mice fed with Control-enriched diet (striped bars) and Bronze-enriched diet (white bars). (**B**) Representative density plot analysis of intracellular staining of MLNs from Winnie mice. (**C**,**D**) Intracellular staining of IFNγ and IL-17A in the CD4^+^ (**C**) and CD4^−^ (**D**) MLN cells of WT and Winnie mice after 2 weeks of Control or Bronze-enriched diets. Statistical evaluation was performed using unpaired two-tailed Student’s *t*-tests. * *P* < 0.05.

**Figure 3 nutrients-10-01862-f003:**
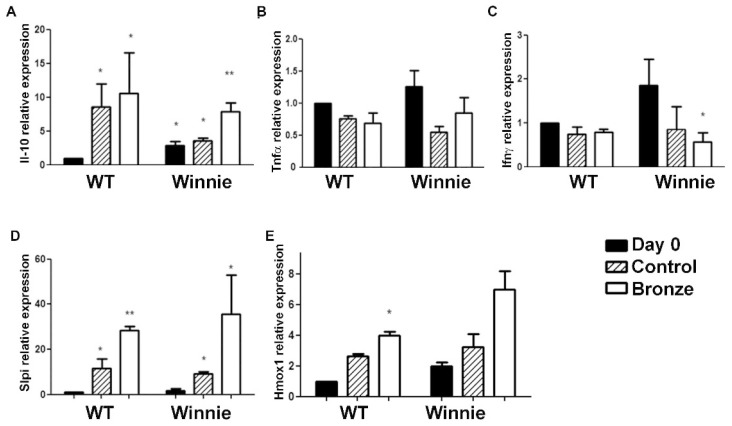
Two weeks of Bronze-enriched diet induced Slpi and Hmox1 expression in the medial colon of WT and Winnie mice. Histograms represent the average expression of Il-10, Tnf-α, Ifnγ, Slpi and Hmox1 (**A**–**E** respectively) measured by real time PCR in the medial colon of WT and Winnie mice after two weeks of Control- (striped bars) and Bronze-enriched diet (white bars). Black bars represent the gene expression at Day 0. All bars represent mean expression ± SEM for each treatment. Control and Bronze dependent expression were compared to WT Day 0 for the statistical evaluation. Student’s *t*-test was used to compare every measurement to the corresponding WT Day 0 and evaluate the significance of the data. ** *P* < 0.01,* *P* < 0.05 (Student’s *t*-test). Grouped analyses were performed with the two-way ANOVA test, using Bonferroni as a post test.

**Figure 4 nutrients-10-01862-f004:**
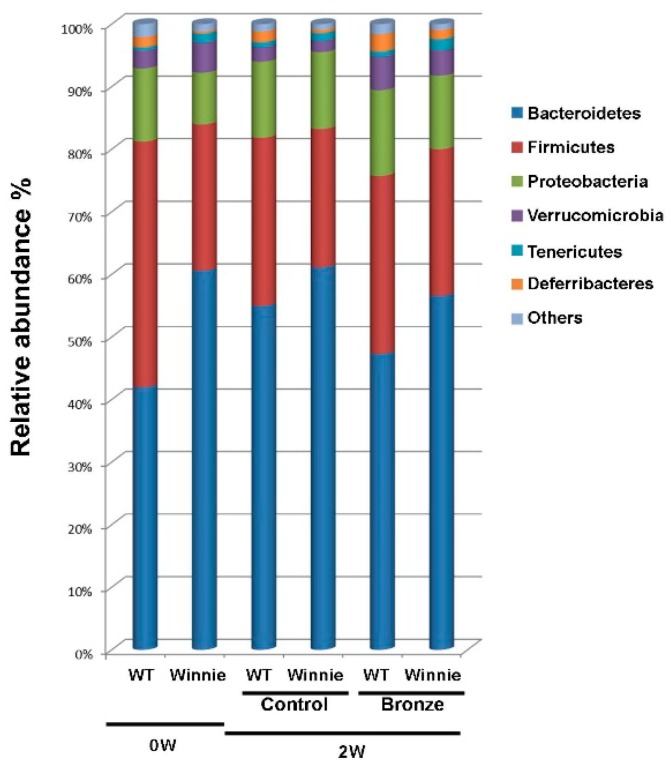
Total bacteria found in feces of WT and Winnie mice. Relative abundance (%) of total (16S rRNA) bacteria, found at the phylum level in the fecal samples of WT and Winnie mice at Day 0 (0W) and after two weeks (2W) on Bronze (Bronze) or Control tomato (Control) diets.

**Figure 5 nutrients-10-01862-f005:**
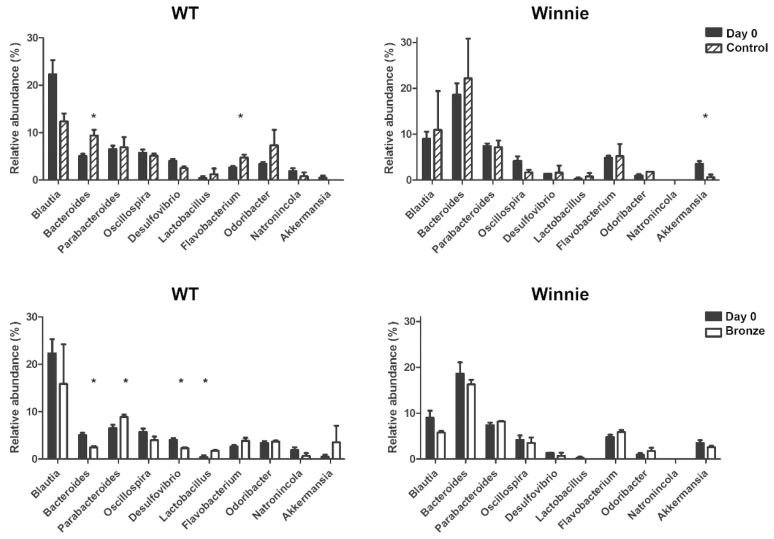
The Bronze-enriched diet affected the fecal bacterial genera of both WT and Winnie mice. Relative abundance (%) of total (16S rRNA) bacteria, found at the genus level in the fecal samples of WT and Winnie mice at T0 (Day 0, black bars) and after two weeks of Bronze (white bars) or Control tomato (striped bars) diet.

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
