# Peer review of "A Bronze-Tomato Enriched Diet Affects the Intestinal Microbiome under Homeostatic and Inflammatory Conditions"

_nutrients, 2018, doi:10.3390/nu10121862_

Round 1
Reviewer 1 Report
In this article, Liso and co-workers investigated the possibility that a bronze-tomatoes enriched dietary regimen could induce a switch in the intestinal microbiota composition in an experimental model of intestinal inflammation. To this end, the authors administered a bronze-tomatoes enriched diet to wild-type (WT) and to Winnie mice that spontaneously develop colitis. In a previous paper, the authors have already demonstrated that bronze tomatoes diet was able to delay intestinal damage induced by DSS.
In this study the authors show that: 1. Administration of a bronze-tomatoes enriched diet do not affect the macroscopic appearance of the colon of both WT and Winnie mice; 2. Bronze tomatoes diet administration for two weeks resulted in a reduction of Il-17A+ CD4 cells in the mesenteric lymph nodes of both WT and Winnie mice; 3. Bronze tomatoes diet is associated with an increase of Slpi and Hmox 1 RNA relative expression in the colon of WT and Winnie mice; 4. Two weeks of bronze tomatoes enriched diet changed the fecal microbiota composition of both WT and Winnie mice with some differences between the two.
I have few suggestions and minor observations:
1. In my opinion it would be appropriate to discuss in the introduction or in the discussion section some of the many recent experimental evidence on the role of the diet in the development and maintenance of intestinal inflammation. This could strengthen the scope of the manuscript and substantiate the findings of the present article.
2. Despite the absence of any significant changes, histological appearance of the colon of WT and Winnie mice should be added to figure 1.
3. Figure legend 3: please specify in the figure legend that cytokines and other factors were measured by real time PCR.
4. In the discussion section (line 2-4), the authors state that anti-TNF-a agents have substituted traditional treatments such as steroids and conventional immunosuppressive drugs in the management of IBD patients. This statement is not accurate, since biological treatments have changed the therapeutic approach, but these drugs have not substituted conventional treatments that are currently used in the acute phases (corticosteroids) or as maintenance therapy (immunosuppressive drugs). Please rephrase.
5. Since, there were no macroscopic changes between WT and Winnie mice after two weeks of the bronze-tomatoes diet, have the authors considered a prolonged duration of the dietary regimen? This aspect should be addressed in the discussion section.
Author Response
We are extremely glad by the reviewers comments and enthusiasm for our work and we would like to use this opportunity to thank them. We revised our manuscript “A Bronze-tomato enriched diet affects the intestinal microbiome under homeostatic and inflammatory conditions.” following their suggestion and we hope that in the present form they will consider it suitable for publication.
In details,
following reviewer 1 suggestions we:
- discussed recent experimental evidence on the role of the diet in the development and maintenance of intestinal inflammation from line 322 to 326.
- included supplementary figure 1 showing intestinal histology.
- explicated that “Il-10, Tnf-α, Ifnγ, Slpi and Hmox1 (A-E respectively) measured by real time PCR in the medial colon of WT and Winnie mice” line 244-245.
- totally agree and thank the reviewer for highlighting our mistake, we changed our sentence from line 310 to line 313.
- agree with the reviewer. Our future plan is to investigate nutritional mediated UC prevention, experiments were already performed and metagenomics data are in our hard drive but not yet analyzed. We discuss this issues in the conclusions from line 382 to 387 and we will be happy to submit our story to this prestigious journal in the next future.
Reviewer 2 Report
Marina Liso et al. beatifully showed the new results supporting their previous findings. I found study sounds clear and good. However, I believe that there are place through the manuscript that can be improved. I cannot classify the changes that I would suggest as minor or major but they are critical for clarity. Here is the points that I need that authors needs to edit:
- Statistical analysis for taxonomy analysis are not well defined. Did you use MaAsLin or any relevant analysis? Please identify in your method parts.
- How many individual experiment performed for each figure? I couldn't see any information through the paper. Even if I just somewhere through the text it is not clear. Please add into the figure legends those information.
- In Fig. 3 there are several stars for statistic. These needs to be shown on figure compared to what these stars are generated.
Author Response
We are extremely glad by the reviewers comments and enthusiasm for our work and we would like to use this opportunity to thank them. We revised our manuscript “A Bronze-tomato enriched diet affects the intestinal microbiome under homeostatic and inflammatory conditions.” following their suggestion and we hope that in the present form they will consider it suitable for publication.
In details,
following reviewer 2 suggestions we:
- revised the material and methods chapter and addressed Reviewer 2 comments from line 131 to 132 and from line 170 to line 177.
- addressed Reviewer 2 question from line 168 to 169 “All results were obtained from three consecutive and independent experiments”.
- better elucidated that all the statistical analysis are compared with WT at day 0, particularly from line 248 to 249.
To address both reviewers request to improve the manuscript English quality, we asked Prof. Cathie Martin to correct our (unfortunately frequent) mistakes.